# Effect of Light-Curable Resin-Modified Glass Ionomer Varnish on Non-Cavitated Proximal Caries Lesions in Primary Molars: A Randomized Controlled Trial

**DOI:** 10.3390/children10071164

**Published:** 2023-07-03

**Authors:** Jihan A. Khan, Najlaa M. Alamoudi, Eman A. El-Ashiry, Osama M. Felemban, Sara M. Bagher

**Affiliations:** 1University Medical Services Center, King Abdulaziz University, Jeddah 21589, Saudi Arabia; jakhan@kau.edu.sa; 2Pediatric Dentistry Department, Faculty of Dentistry, King Abdulaziz University, Jeddah 21589, Saudi Arabia; nalamoudi@kau.edu.sa (N.M.A.); eelashiry@kau.edu.sa (E.A.E.-A.); sbagher@kau.edu.sa (S.M.B.)

**Keywords:** light-curable resin-modified glass ionomer varnish, non-cavitated proximal carious lesions, Vanish XT varnish

## Abstract

The purpose of this study was to radiographically and clinically assess, after six and twelve months, the additive effect of light-curable resin-modified glass ionomer (LCRMGI) varnish on preventive standard-of-care measures (sodium fluoride varnish, oral hygiene instructions, and dietary counseling) for the treatment of non-cavitated proximal carious lesions in primary molars. This prospective, split-mouth, randomized clinical trial recruited 5–8-year-old children with at least one pair of bilateral non-cavitated proximal carious lesions in the enamel or outer third of the dentin. The lesions were randomized into one of two groups: experimental (which received LCRMGI varnish with the preventive standard-of-care measures) or control (which received the preventive standard-of-care measure). A total of 47 out of 53 participants, with 70 out of 80 pairs, completed the study. Radiographically, lesions in the experimental groups were more likely to regress after six (OR = 3.25) and twelve months (OR = 2.67), but it was not statistically significant (*p* = 0.052 and *p* = 0.055, respectively). Clinically, the experimental group showed significantly higher regression rates than the control group after six and twelve months (*p* = 0.041 and *p* = 0.003, respectively). The LCRMGI varnish demonstrated promising potential as an adjuvant to preventive standard-of-care measures in promoting regression and inhibiting the progression of non-cavitated proximal carious lesions.

## 1. Introduction

Clinically, carious lesions can be categorized as cavitated or non-cavitated lesions. Traditional non-invasive preventive approaches for the treatment of non-cavitated carious lesions aim to control the biofilm through provisionally applied remineralization agents, antibacterial agents, or mechanical plaque removal by the patient [1]. However, their success is often affected by the patient’s compliance [2]. The more invasive restorative approach, on the other hand, necessitates the removal of the sound tooth structure to a great extent, especially for proximal carious lesions, even when minimally invasive procedures are used [3]. Therefore, micro-invasive procedures have been developed as a middle ground between the other two sides of treatment [4]. Evidence from limited research in permanent and primary teeth has revealed that micro-invasive therapies are generally more effective than non-invasive techniques (such as professionally applying fluoride varnish) in arresting non-cavitated proximal carious lesions in the enamel and outer dentin [1,3]. Furthermore, the similar procedure time of some micro-invasive techniques, such as with the resin infiltration (RI) technique, to that of classic restorations, as well as the higher cost in comparison to the standard preventive methods, urges the exploration of other alternative options for the management of lesions on the proximal surfaces that are non-cavitated [5,6].

Light-curable resin-modified glass ionomer (LCRMGI) varnishes have been created as localized protective varnishes for particular sites on dentin and enamel tooth surfaces due to their gradual fluoride release properties that last for up to six months [7]. Several randomized clinical trials were carried out and reported that the use of LCRMGI varnishes demonstrated satisfactory results in minimizing hypersensitivity in the dentin [8] for up to six months after treatment [9], preventing enamel demineralization during comprehensive orthodontic treatment [10,11], and as a sealant in preventing caries [12]. There is a lack of studies on the effect of these LCRMGI varnishes on controlling the progression of non-cavitated proximal carious lesions. This emphasizes the need to conduct such research, particularly in the young pediatric population, due to the material’s unique features, non-invasive nature, ease of application, and economic benefits.

The purpose of this study was to compare the additive effect of the LCRMGI varnish with the preventive standard-of-care measures (an application of 5% of sodium fluoride varnish (NaF), oral hygiene instructions (OHI), and dietary counseling) to the preventive standard-of-care measures alone, on promoting the regression or inhibiting the progression of non-cavitated enamel and outer dentine proximal carious lesions in primary molars evaluated by bitewing (BW) radiographs and direct visual clinical examination after six and twelve months. We hypothesized that the success rate of the LCRMGI varnish (with the preventive standard-of-care measures) would be higher than that of the preventive standard-of-care measures alone.

## 2. Materials and Methods

The study was a blinded, randomized, controlled clinical trial with a split-mouth design and a follow-up period of up to 12 months. The reporting of the study followed the Consolidated Standards of Reporting Trials (CONSORT) guidelines for reporting randomized trials [13]. The study was approved by the Research Ethics Committee at King Abdulaziz University, Faculty of Dentistry, Jeddah, Saudi Arabia. The study protocol is registered at www.clinicaltrials.gov (accessed on 4 December 2018) under the identifier NCT03685058.

### 2.1. Eligibility Screening

Children who attended the Pediatric Dental Clinics in King Abdulaziz University Dental Hospital (KAUDH) between December 2017 and February 2019 for primary examination or routine follow-up were screened for eligibility to be included in the study. The screening was composed of three phases. The screening criteria of the first phase was to identify children who were five to eight years old, healthy, with the American Society of Anesthesiologists (ASA) classification of ASA I, not known to be allergic to any of the dental materials’ components that would be utilized in the study, generally cooperative for dental treatment with a Frankl Behavioral Rating Scale rating of “definitely positive” or “positive” [14], and had no history of mouth breathing or dry mouth. The participants must also have had at least one pair of matched bilateral primary molars in contralateral quadrants that were free of anatomical abnormalities, were not mobile, did not require a restorative treatment, and were without frank cavitation or previous restoration.

Children who met the screening inclusion criteria in the first phase qualified for the second phase of the screening. The study aim was introduced to the parent/guardian of potentially eligible children, and verbal consent was obtained from them to take standardized screening BW radiographs. In the second phase, qualified children were screened radiographically for enamel and outer dentine proximal carious lesions by utilizing standardized BW radiographs. To facilitate a standard and reproducible X-ray projection geometry for the follow-up visits’ BW radiographs, all the radiographs were taken by the primary investigator (PI), and the central beam was pointed toward the interproximal area between the mandibular first primary and second primary molars while using the extension cone paralleling (XCP) Rinn holder and X-ray beam centering system for accurate photo-stimulable phosphor plate positioning. The exposure time was set at 0.05 s and the voltage at 70 KV, 7 mA. The screening standardized BW radiographs were coded and then graded independently by two trained and calibrated examiners (J.K. and O.F.) on a 19-inch screen in a dark room. In the case of disagreement between the first two examiners (J.K. and O.F.), a third trained and calibrated examiner (S.B.) was asked to evaluate the radiographs independently. The Kappa’s values for the inter-examiner reliability (0.847) and intra-examiner reliability (0.847–0.850) for the radiographic examination indicated very good agreement between the examiners.

The following radiographic grading system was utilized [15]: (0) No radiolucency; (E1) radiolucency within enamel outer half; (E2) radiolucency in enamel inner half; (D1) radiolucency in dentin outer third; (D2) radiolucency in dentin middle third; (D3) radiolucency in dentin inner third. Any child diagnosed radiographically with a proximal carious lesion with a score of D2 or more was considered ineligible, and an appointment was scheduled for restorative treatment. If a child was diagnosed by at least two of the examiners to have at least one pair of primary molars in contralateral quadrants with matching bilateral proximal carious lesions (E1, E2, and D1) he/she was eligible for the third screening phase, which included clinical verification of the cavitation status of the lesion.

The third screening phase was comprised of a direct visual clinical examination to assess the cavitation status of the proximal carious lesions that were deemed qualified in the second phase. To enable direct visual clinical examination of the proximal surfaces, orthodontic elastic separators were placed in the eligible interproximal areas. The children returned after two days, and the temporary separators were removed. The selected proximal surfaces were cleaned using a prophy cup (Prophy Cup, Snap-on, Guangzhou Jaan Medical CO., Guangzhou, Baiyun, China) and dental floss (Dental Floss, non-waxed, Oral-B Laboratories, Iowa City, IA, USA). The surfaces were examined by the gentle running of a community periodontal index (CPI) probe across the lesion to assess cavitation. Figure 1 shows an illustration of the clinical examination steps. Then, carious lesions were clinically scored, from the lingual, buccal, and occlusal sides, with the International Caries Detection and Assessment System II (ICDAS II) [16], with a score of (0) indicating clinically sound surfaces; (1 and 2) were non-cavitated lesions with score 1 lesions requiring air-drying to see the lesion, while the lesion could be seen without air-drying in score 2; (3) there was a localized breakdown in the enamel because of caries, but the dentin was not visible; (4) had a dark underlying dentin shadow with or without a localized enamel breakdown; (5 and 6) had dentine cavitation with increasing stages. The surfaces were scored by the two trained and calibrated examiners independently (.JK. and O.F.). If there was disagreement in the scores between the first two examiners, a third (S.B.) calibrated and trained examiner was asked to assess the lesion. The Kappa’s values for the inter-examiner (0.869) and intra-examiner reliability (0.867–0.870) for the direct visual clinical examination indicated very good agreement between the examiners. During the clinical examination, all examiners were blinded to the radiographic score of the lesions. Any child diagnosed with a cavitated proximal carious lesion (ICDAS II score of 3 or more) by at least two of the examiners was excluded and referred for restorative treatment. Children with at least one pair of primary molars in contralateral quadrants with matching non-cavitated proximal carious lesions diagnosed radiographically as E1, E2, and D1 and clinically as an ICDAS II score of 1 or 2 were considered eligible for the study.

### 2.2. Study Procedures

Parents of participants who agreed to participate in the study signed a written Arabic consent form following a thorough description of the treatment process, including potential outcomes, advantages, drawbacks, and risks. In addition to the baseline standardized BW radiographic and direct visual clinical examination scores, the medical history, age, gender, nationality, caries risk assessment (CRA), the decayed, missing, and filled due to the caries index for permanent (DMFT) teeth, the decayed, missing, and filled due to the caries index for primary (dmft) teeth, and the simplified debris index (DI-S) component of the simplified oral hygiene index (OHI-S) [17,18,19] of the included participants were recorded.

For the experimental group, the lesions were treated with LCRMGI Vanish^TM^ XT varnish (Vanish™ XT Extended Contact Varnish, 3M ESPE, St. Paul, MN, U.S.A.) by a single operator, J.K. A few drops of local anesthesia, lidocaine HCl 2% with epinephrine 1:100.000 (Octocaine^®^ 100, Novocol Healthcare Inc. Cambridge, ON, Canada), was used to anesthetize the free gingiva before rubber dam isolation. A celluloid strip and a wedge were applied adjacent to the lesion surface utilizing the spaces created by the orthodontic separators. The Vanish^TM^ XT was applied following the instructions of the manufacturer. After the tooth surface was cleaned with the prophy cup and floss, it was etched using 35% phosphoric acid (H_3_PO_4_) (Scotchbond Universal Etchant, 3M ESPE, St Paul, MN, USA) for 15 s (maximum etching time was 60 s); then, it was rinsed with water, and excess pooled water was air dried. The liquid/paste materials were mixed for 10–15 s until a smooth consistency and glossy appearance were reached. The material was applied on the tooth surface in a thin layer (0.5 mm or less), and the material was light-cured for 20 s. The surface was then flossed and checked for any excess material, and finally, a moistened cotton applicator was used to wipe the coating to remove the thin film on the surface.

For the control group lesions, the 5% NaF varnish (Vanish^TM^ 5% Sodium Fluoride White Varnish with Tri-Calcium Phosphate, 3M ESPE, St. Paul, MN, USA) was applied to the control’s tooth surface, then to all the rest of the teeth (including the experimental group lesions). Other preventive measures, including the OHI, daily flossing, and diet counseling, were given to the participants.

### 2.3. Randomization

With a block size of four, block randomization was applied. Since a split-mouth technique was used, it was determined that the right side would always receive the allocation from the randomization table, and the left side would receive the alternative treatment. Even if the participant had multiple included pairs, all of the lesions on the right side of the dental arch received the treatment assigned in the table, and all of the lesions on the left side received the alternative treatment. For allocation concealment from the operator, the randomization sequence was kept with a dental staff member. Each time a participant was included, the designated assistant would advise the operator of the participant’s treatment allocations.

### 2.4. Follow-Up Visits

Follow-up visits took place after six and twelve months. New standardized BW radiographs and direct visual clinical examination (after temporary teeth separation) were performed using the same methods and equipment as the baseline and were scored by the same two trained and calibrated examiners independently. All the examiners were blinded to the group allocation of each lesion and the baseline radiographic and clinical scores of each lesion. Any lesion that was clinically cavitated at the follow-up visits was restored.

At each follow-up visit, the participants’ medical history and CRA were also reviewed, and the OHI and diet counseling were reinforced to both participants and their parents. The LCRMGI varnish was re-applied in the experimental group lesions in the same technique it was applied during the baseline application. All the teeth in both groups (experimental and control) received the application of 5% NaF varnish.

### 2.5. Statisitcal Analysis and Sample Size

The demographic information, the individuals’ oral health condition, the locations of the lesion pairs, and included teeth were all described using univariate analysis. Using chi-square or Fisher’s exact tests, the ICDAS II and radiographic diagnostic scores were independently compared between the groups at the baseline, six and twelve months.

Each lesion was classified as progressing if the score increased, being arrested if it remained the same score, or regressing if the score decreased by comparing the six months scores to the baseline scores and the twelve months scores to the six months scores. The experimental group’s rates of progression, arrest, and regression at the 6- and 12-month follow-up visits were compared to those of the control group using the chi-square or Fisher’s exact test. For lesions with clinical cavitation at the 6-month follow-up visit that were restored (according to the study protocol), the carryover method was used, i.e., the same score was recorded in the 12-month follow-up visit by the ICDAS II and the radiographic examination. This was used to retain the other lesion in the pair in the study and keep monitoring it for caries progression.

The possible within-participants effect brought on by the split-mouth design was taken into consideration in the final analysis. Each matched pair of lesions was assigned to one of four categories: both lesions successful, experimental successful and control failed, experimental failed and control successful, or both lesions failed. At one point, success was defined as regression or arrest and failure as progression; then, success was redefined as regression, and failure was redefined as arrest or progression. The proportions of the paired data between the experimental and control groups were examined using a McNemar test. The statistical threshold for significance was set at *p* < 0.05. Data analysis was carried out using IBM SPSS Statistics for Windows, Version 20.0.

For the sample size calculation, when the proportion of discordant pairs was predicted to be 31%, and the technique of analysis was a McNemar test of equality of paired proportions with a 5% two-sided significance level, a sample size of 69 pairs of lesions would have 80% power to detect a difference in proportions of 18%. The sample size was increased by 15% to account for loss to follow-up or other attrition causes. Thus, a total of 80 pairs, or 160 lesions, were needed at the start of the clinical trial. The sample size was determined utilizing the program nQuery (2017) Sample Size and Power Calculation (Statistical Solutions Ltd., Cork, Ireland).

## 3. Results

A total of 357 of the 1187 five- to eight-year-old children who visited the pediatric dentistry clinics between December 2017 and February 2019 at KAUDH were qualified for the study’s screening process. The study’s inclusion criteria were met by 53 of them, and their parents/guardians approved of their participation. The participants had a total of 80 matched pairs of non-cavitated proximal carious lesions in their primary molars, which amounted to 160 lesions. Three participants (five pairs) dropped out of the research at the six-month follow-up visit (dropout rate: 5.7% participants; 6.3% pairs). An additional three participants (five pairs) dropped out of the research at the 12-month follow-up visit (total dropout rate: 11.4% participants; 12.6% pairs). The CONSORT flow chart for the participants up to the 12-month follow-up visit is shown in Figure 2.

Table 1 lists the demographic characteristics of the study participants at the baseline. The caries risk was high for all the included participants due to the presence of more than one proximal carious lesion.

Table 2 demonstrates the baseline characteristics of the included lesion pairs. The included pairs were evenly split between the mandibular and maxillary arches. The upper second primary molar (36.3%) and lower second primary molar (35.0%) were the most commonly recruited pairs in this study. Most of the included lesions were located on the mesial surface (61.3%).

Table 3 shows the lesion characteristics by clinical and radiographic examination scores at the baseline and the 6- and 12-month follow-up visits in the experimental and control groups. Regarding the results of the ICDAS II clinical diagnosis, the majority of the lesions scored 2 (non-cavitated lesions seen without air-drying) in both the experimental and control groups at the baseline (83.8% and 87.5%, respectively) and the 6-month (64.0% and 78.7%, respectively) and 12-month follow-up visits (58.6% and 62.9%, respectively). There was no statistically significant difference in the ICDAS II scores between the two groups at the baseline (*p* = 0.499) or at the six-month follow-up visit (*p* = 0.073). But a statistically significant difference in the ICDAS II scores between the two groups at the 12-month follow-up visit (*p* = 0.047) was recorded, with more lesions in the experimental group having scores of 0 (5.7%) and 1 (17.1%) than in the control group (1.4% scored 0, and 5.7% scored 1).

In terms of lesion cavitation, ten lesions (13.4%) (eight scored 3 (10.7%), two lesions scored 4 (2.7%)) in the experimental group, and nine lesions (12.0%) in the control group exhibited clinical cavitation and required restorations at the six-month follow-up visit. However, at the 12-month follow-up visit, twelve lesions (17.2%) in the control group were clinically cavitated (nine (12.9%) scored 3, two (2.9%) scored 4, and one (1.4%) scored 5 and 6), compared to just three lesions (4.3%) that scored 3 in the experimental group. A clinical examination example of assessing the lesions is shown in Figure 3.

The findings of the radiographic scores obtained by both the experimental and control groups at the baseline and the 6- and 12-month follow-up visits exhibited a tendency that was comparable to that observed in the ICDAS II; nevertheless, there was no statistically significant difference between the two groups in the radiographic scores at the baseline and follow-up visits. Figure 4 is an illustration of the lesion evaluation using radiographs.

Table 4 shows the incidence of clinical and radiographic regression, arrest, or the progression of the lesions from the baseline through the 6- and 12-month follow-up visits. Clinical examination revealed that most of the lesions were arrested at the 6-month follow-up visit in the experimental and control groups (73.3% and 81.3%, respectively) and the 12-month follow-up visit (60.0% and 62.9%, respectively). Although the results for the frequency of clinical regression, arrest, and progression of the lesions at the 6-month follow-up visit between the groups were not significantly different (*p* = 0.166), they were statistically significantly different at the 12-month follow-up visit (*p* = 0.003). The radiographic examination showed that most of the lesions were arrested at the 6-month follow-up visit in the experimental and control groups (64% and 70.7%, respectively) and the 12-month follow-up visit (45.7% and 48.6%, respectively). In addition, more lesions progressed in the control group at the 6-and 12-month follow-up visits (22.7% and 35.7%, respectively) than in the experimental group (17.3% and 24.3%, respectively). Overall, the incidence of radiographic regression, arrest, and progression of the lesions from the baseline to the 6- and 12-month follow-up visits was not significantly different between the groups (*p* = 0.080 and 0.095, respectively).

To account for confounding variables, Table 5 displays the clinical and radiographic success (the regression or arrest of the carious lesions) vs. failure (the progression of the carious lesions) outcomes for each lesion pair. Most of the lesions were successful clinically in both groups in the 6- and 12-month follow-up visits (74.7%, and 55.7%, respectively). The lesions in the experimental group had 2.5 times the odds (95% CI, 0.92–7.87) of clinical success (regression or arrest) after 12 months compared to lesions in the control group, but the difference was not statistically significant (*p* = 0.081). The findings of the radiographic examination revealed that most of the lesions were also successful radiographically in both groups at the 6-month follow-up visit (62.75%). In the 12-month follow-up visit, only half of the lesions were successful radiographically in both groups (50.0%), and the number of pairs where the experimental lesions succeeded and the control lesions failed in the 12-month follow-up visit was double (n = 18), the number of pairs in which the experimental lesions failed, but the control lesions succeeded (n = 9), the difference was not statistically significant (*p* = 0.124).

Table 6 shows the outcomes of the success and failure in each paired lesion when the definitions of success and failure were altered to success, defined as the regression of the carious lesion, and failure, defined as the arrest or progression of the carious lesion. The ICDAS II clinical evaluation revealed that the experimental treatment’s clinical ability to produce caries regression was significantly higher than the control after the 6- (*p* = 0.041) and 12-month follow-up visits (*p* = 0.003), even though most of the lesions failed in both groups at the 6- and 12-month follow-up visits (89.3%, and 82.9%, respectively). The radiographic examination also revealed a greater number of pairs in which the experimental group lesions succeeded and the control failed at 6 (17.3%) and 12 months (22.9%) compared to pairs in which the experimental treatment failed, and the control succeeded (5.3%, and 8.6%, respectively). In other words, lesions in the experimental group had 3.25 (95% CI, 1.0–13.7) odds of regression after 6 months and 2.67 (95% CI, 0.99–8.32) odds of regression after 12 months compared to lesions in the control group, but the results were not statistically significant (*p* = 0.052 and *p* = 0.055, respectively).

## 4. Discussion

The results from the present study showed that the addition of LCRMGI varnish to the preventive standard-of-care measures enhanced the regression of non-cavitated proximal carious lesions to a certain extent in primary molars after 6 and 12 months of treatment. The preventive standard-of-care measures (using professionally applied NaF, the OHI, and dietary counseling) were used as the control treatment because it is regarded as the gold standard non-invasive treatment for non-cavitated carious lesions [20]. However, it has a drawback in its reliance on the compliance of the patient in committing to the topical fluoride application visits and in following the oral hygiene and dietary instructions [1].

LCRMGI varnish was used (in addition to the preventive standard-of-care measures) as the experimental material as an innovative approach for the micro-invasive management of non-cavitated proximal carious lesions. It is a light-curable RMGI varnish and has the properties of glass ionomer materials, such as bonding and remineralization, by constantly discharging and uptaking fluoride for a prolonged time [7]. It also contains calcium glycerophosphate, which supplies phosphate and calcium [21]. In addition, in comparison to the RI micro-invasive method, it is easier and faster to apply and more economical. It has been used successfully in other clinical applications, mainly in caries prevention [10,11,12] and dentinal hypersensitivity treatment [8,9].

To achieve a comprehensive and detailed evaluation in monitoring the status of the carious lesions during the course of the research, both radiographic and direct clinical examination after temporary teeth separation diagnostic methods were used to monitor the lesions in the study. Firstly, BW radiographs were used, as they are the traditional method to detect carious lesions on posterior proximal surfaces in support of the visual-tactile examination [22]. Secondly, the direct clinical examination method after temporary teeth separation was used to ensure that the included lesions were non-cavitated to avoid any uncertainties on whether the cause of failure of the treatment was due to undetected cavitation of the lesion from the start or because of the real failure of the treatment [23]. It also provided the space that facilitated the direct application of the LCRMGI varnish on the surface of the lesion at the same time.

In the results of the present study, 88.7% of the participants were available after 12 months of treatment. The participants lost to follow-ups might have been attributed to the need for two appointments for the temporary teeth separation at the follow-up visits, which was not convenient for some of the participants and/or their parents.

At the 6- and 12-month follow-up visits, there was no significant difference in the lesion characteristics by radiographic examination. Nevertheless, there was a statistically significant difference in the ICDAS II scores between the two groups. More lesions with scores of 0 and 1 were seen in the experimental group, and more lesions with a score of 2 and higher were seen in the control group. The significant shift of the difference in the distribution of the clinical scores between the two groups after 12 months might be related to the added benefit from the LCRMGI varnish treatment to the preventive standard-of-care measures in the experimental group.

It was also decided to view the success of the treatments in two ways; the first was lesion regression, and the second was lesion regression or arrest (i.e., no progression). This was because of the nature of the experimental and control group materials that were used. They both contain remineralizing agents (fluoride, calcium, and phosphate) but in different systems. Thus, it was of interest to analyze the difference between the two materials in the extent of their remineralizing potential in the clinical setting by measuring the success of each treatment in promoting the ultimate outcome of lesion regression, in addition to measuring their success in inhibiting lesion progression in general (arrest and regression).

Although the results showed more lesion regression clinically and radiographically in the experimental group at the 6- and 12-month follow-up visits, it was only significant clinically at the 12-month follow-up visit. These results agreed with another split-mouth RCT that used a GIC sealant to treat initial proximal carious lesions in permanent posterior teeth. After 12 months, significantly more lesions treated with GIC in their study regressed radiographically compared to the controls [24].

With regards to lesion progression, higher but not significant lesion progression frequency was seen in the control than in the experimental group at both the 6- and 12-month follow-up visits radiographically and at the 12-month follow-up visit clinically. This agrees with another split-mouth micro-invasive intervention study using RI in primary molars, in which higher but not significant lesion progression radiographically at their 6- and 12-month follow-ups visits was also reported in the control group (treated with the preventive standard-of-care measures) than those in the case group (treated with RI and the preventive standard-of-care measures) [2]. In addition, another study reported significantly more lesion progression radiographically in the control group only (treated with fluoride varnish) than those that showed progression in only the experimental group (treated with RI with fluoride varnish) after one year of treatment [25].

With regards to the frequency of lesion arrest, it was found that the highest percentage of the lesions were arrested by clinical and radiographic examination in both the experimental and control groups during the duration of the study. This was also seen when the success of the lesion pairs was described as regression or arrest of the carious lesions, where most of the lesions were successful clinically in both groups at the 6- and 12-month follow-up visits and radiographically at the 6-month follow-up visit, while half of the lesions were successful radiographically in both groups at the 12-month follow-up visit.

To the best of our knowledge, the present study is considered one of the initial studies examining the success of this LCRMGI varnish in the treatment of non-cavitated proximal carious lesions. Therefore, it was only possible to compare the results of the present study to studies that similarly applied other types of micro-invasive interventions for the treatment of non-cavitated proximal carious lesions, such as GIC sealants [24], sealing with adhesives [26], and RI [2,25]. It is essential to note that the difference in the results between the LCRMGI varnish used in the present study and RI could be related to the difference in the nature of the materials and their mechanisms of action, which should be considered when comparing their results. The LCRMGI varnish aims to provide a remineralizing coating over the surfaces of non-cavitated carious lesions that could extend for a prolonged time [7,27]. RI, on the other hand, is a low-viscosity light-curable resin that intends to create a diffusion barrier within the lesion to block the diffusion of the dissolved minerals and acids by soaking the porous body of the enamel lesion with the low-viscosity infiltrant followed by light curing. It thus mainly aims towards the arrestment of the progression of non-cavitated carious lesions.

Another factor that should also be considered when comparing the findings of the present study to some of the other micro-invasive intervention studies [2,24] is that such studies included only lesions in the enamel (by radiographic examination), but in the current study, lesions radiographically reaching the outer dentin were also included; at the same time, a direct visual examination was performed after temporary teeth separation to ensure that all the included lesions in the present study were non-cavitated and eligible for non-invasive and micro-invasive treatments.

The study had several limitations: the need for two appointments for the temporary teeth separation and the occasionally associated discomfort with the orthodontic separator may have contributed to the attrition of participants at the follow-up visits. Also, due to the nature of the treatments, it was difficult to blind the operator to the type of treatment each lesion received.

In view of the limitations of this research, further studies are recommended and longer follow-ups to reach evidence-based recommendations. Also, studies comparing the LCRMGI varnish with other types of micro-invasive interventions in controlling the progression of non-cavitated carious lesions on proximal surfaces in primary molars in children are suggested. Furthermore, it is also recommended to develop a special innovated application apparatus to facilitate the application of LCRMGI varnishes to the proximal surfaces without the need for temporary teeth separation.

## 5. Conclusions

Based on the results of the present study, it can be concluded that LCRMGI varnishes with preventive standard-of-care measures showed significantly higher odds for promoting the regression of non-cavitated proximal carious lesions by clinical assessment than the preventive standard-of-care measures alone after 6- and 12-months of treatment. This was also seen by radiographic assessment, but it was not statistically significant.

## Figures and Tables

**Figure 1 children-10-01164-f001:**
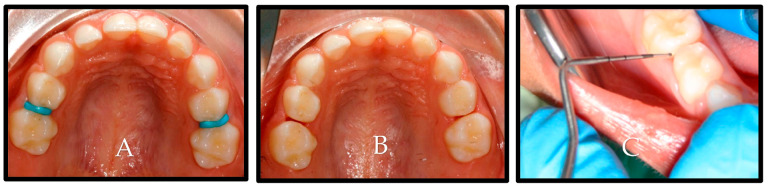
Clinical examination steps. (**A**) Separators in place for two days.; (**B**) separators removed after two days, and space created; and (**C**) CPI probe was used to verify cavitation.

**Figure 2 children-10-01164-f002:**
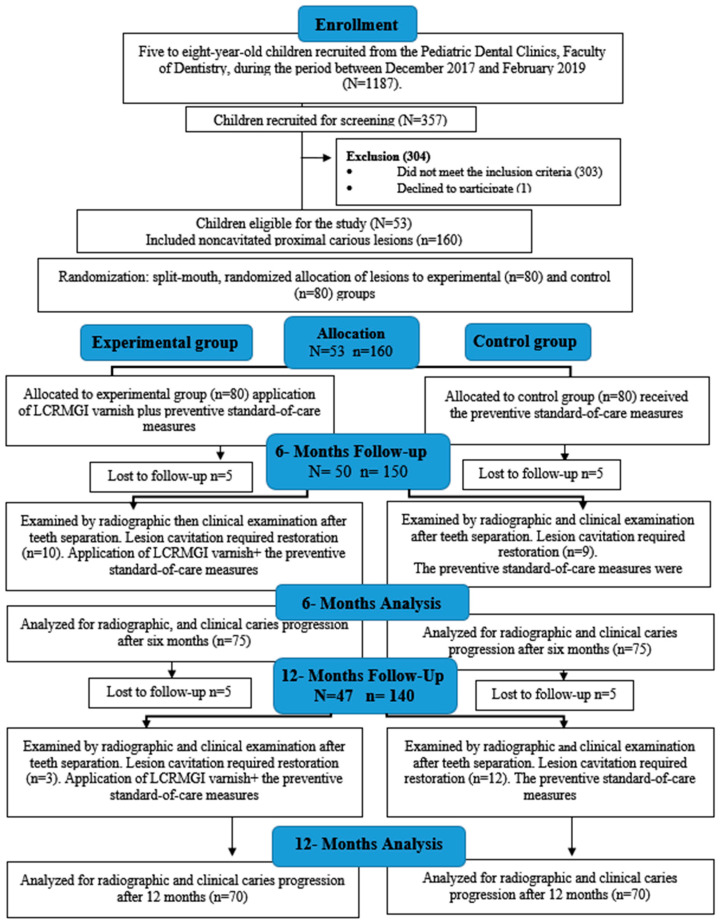
Study flow diagram up to 12-months follow-up. N: is for the number of participants, and n is for the number of non-cavitated proximal carious lesions.

**Figure 3 children-10-01164-f003:**
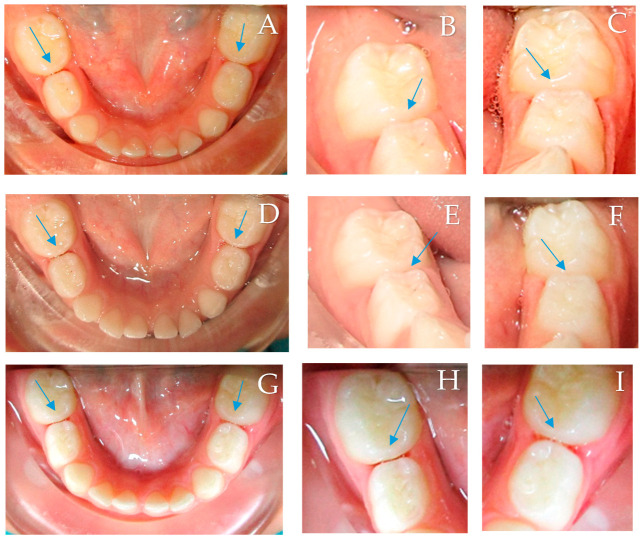
Clinical examination of the same patient at the baseline after 6 months and after 12 months. (**A**): Intraoral photographs at the baseline after temporary tooth separation. (**B**): #85 (control) at the baseline mesial carious lesion could be seen only after drying. (**C**): #75 (experimental) at the baseline mesial carious lesion could be seen when wet. (**D**): Intraoral photographs after 6 months after temporary tooth separation. (**E**): #85 (control) after 6 months mesial carious lesion progressed and could be seen when wet. (**F**): #75 (experimental) after 6 months mesial carious lesion regressed and could be seen only when dry. (**G**): Intraoral photographs after 12 months after temporary tooth separation. (**H**): #85 (control) after 12 months mesial carious lesion progressed and could be seen when wet. (**I**): #75 (experimental) after 12 months mesial carious lesion regressed and could be seen only when dry.

**Figure 4 children-10-01164-f004:**
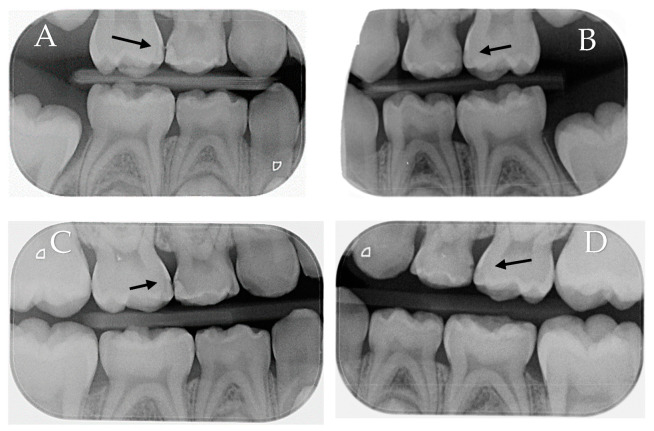
Bitewing radiographs of the same patient at the baseline and after 12 months. (**A**): #55 (control) at the baseline mesial carious lesion (E1). (**B**): #65 (experimental) at the baseline mesial carious lesion (E2). (**C**): #55 (control) after 12 months mesial carious lesion showing slight progression (D1). (**D**): #65 (experimental) after 12 months mesial carious lesion without progression (E2).

**Table 1 children-10-01164-t001:** Demographic characteristics of the study participants at the baseline (N = 53).

Demographics	n (%)
Gender	
Male	23 (43.4)
Female	30 (56.6)
Age in years	
5–6	37 (71.8)
7–8	16 (30.2)
	**Mean ± SD**
Mean age in years	6.00 ± 1.11
Oral health	
dmft	8.04 ± 2.41
DMFT	0.49 ± 1.12
DI-S	1.37 ± 0.52

dmft: Decayed, missing, filled due to caries index for primary teeth. DMFT: Decayed, missing, filled due to caries index for permanent teeth. DI-S: The simplified debris index.

**Table 2 children-10-01164-t002:** Distribution of the included pairs (N = 80 pairs).

Pairs Characteristics	Categories	n (%)
Arch	Maxillary	40 (50.0)
	Mandibular	40 (50.0)
Primary Molar	Upper second primary molar	29 (36.3)
	Lower second primary molar	28 (35.0)
	Upper first primary molar	11 (13.8)
	Lower first primary molar	12 (15.0)
Surface	Mesial	49 (61.3)
	Distal	31 (38.8)

**Table 3 children-10-01164-t003:** Lesion characteristics by radiographic and clinical examination scores at the baseline and the 6- and 12-month follow-up visits in the experimental and control groups (N = 160 lesion).

Diagnostic Method	Baseline		Six Months		12 Months	
	Experimental N = 80 n (%)	Control N = 80 n (%)	Experimental N = 75 n (%)	Control N = 75 n (%)	Experimental N = 70 n (%)	Control N = 70 n (%)
**Clinical (ICDAS II scores)**						
0: Sound	0 (0.0)	0 (0.0)	2 (2.7)	0 (0.0)	4 (5.7)	1 (1.4)
1: Caries seen with drying	13 (16.3)	10 (12.5)	15 (20.0)	7 (9.3)	12 (17.1)	4 (5.7)
2: Caries seen when wet	67 (83.8)	70 (87.5)	48 (64.0)	59 (78.7)	41 (58.6)	44 (62.9)
3: Enamel cavitation	0 (0.0)	0 (0.0)	8 (10.7)	9 (12.0)	3 (4.3)	9 (12.9)
4: Dark dentin	0 (0.0)	0 (0.0)	2 (2.7)	0 (0.0)	0 (0.0)	2 (2.9)
5 and 6: Increasing stages of dentine cavitation	0 (0.0)	0 (0.0)	0 (0.0)	0 (0.0)	0 (0.0)	1 (1.4)
Restoration	0 (0.0)	0 (0.0)	0 (0.0)	0 (0.0)	10 (14.3)	9 (12.9)
*p*-value	0.499 ^†^		0.073 ^‡^		0.047 * ^‡^	
**Radiographic scores**						
0: Sound	0 (0.0)	0 (0.0)	2 (2.7)	0 (0.0)	5 (7.1)	2 (2.9)
E1: Radiolucency in enamel outer 1/2	7 (8.8)	15 (18.8)	12 (16.0)	9 (12.0)	14 (20.0)	6 (8.6)
E2: Radiolucency in enamel inner 1/2	40 (50.0)	29 (36.3)	30 (40.0)	29 (38.7)	25 (35.7)	28 (40.0)
D1: Radiolucency in dentin outer 1/3	33 (41.3)	36 (45.0)	27 (36.0)	34 (45.3)	15 (21.4)	19 (27.1)
D2: Radiolucency in dentin middle 1/3	0 (0.0)	0 (0.0)	4 (5.3)	3 (4.0)	1 (1.4)	4 (5.7)
D3: Radiolucency in dentin inner 1/3	0 (0.0)	0 (0.0)	0 (0.0)	0 (0.0)	0 (0.0)	2 (2.9)
Restoration	0 (0.0)	0 (0.0)	0 (0.0)	0 (0.0)	10 (14.3)	9 (12.9)
*p*-value	0.091 ^†^		0.594 ^‡^		0.191 ^‡^	

^†^ chi-square test, ^‡^ Fisher’s exact test, * Statistically significant (*p* < 0.05).

**Table 4 children-10-01164-t004:** The frequency of clinical and radiographic regression, arrest, and progression of the lesions from the baseline to 6- and 12-month follow-up visits.

Diagnostic Method	Six Months		12 Months	
	Experimental N = 75 n (%)	Control N = 75 n (%)	Experimental N = 70 n (%)	Control N = 70 n (%)
**Clinical (ICDAS II)**				
Regression of the carious lesion	8 (10.7)	2 (2.7)	12 (17.1)	1 (1.4)
Arrest of the carious lesion	55 (73.3)	61 (81.3)	42 (60.0)	44 (62.9)
Progression of the carious lesion	12 (16.0)	12 (16.0)	16 (22.9)	25 (35.7)
*p*-value	0.166 ^‡^		0.003 * ^‡^	
**Radiographic**				
Regression of the carious lesion	14 (18.7)	5 (6.7)	21 (30.0)	11 (15.7)
Arrest of the carious lesion	48 (64.0)	53 (70.7)	32 (45.7)	34 (48.6)
Progression of the carious lesion	13 (17.3)	17 (22.7)	17 (24.3)	25 (35.7)
*p*-value	0.080 ^†^		0.095 ^†^	

^†^ chi-square test, ^‡^ Fisher’s exact test, * Statistically significant (*p* < 0.05).

**Table 5 children-10-01164-t005:** The clinical and radiographic success (regression or arrest of the carious lesion) vs. failure (progression of the carious lesion) outcomes for each pair at the 6- and 12-month follow-up visits.

Diagnostic Method	The Success and Failure Status in Each Pair of Lesions		Six Months N = 75 Pairs	12 Months N = 70 Pairs
	Experimental	Control	n (%) Pairs	n (%) Pairs
**Clinical (ICDAS II)**	Regression or arrest	Regression or arrest	56 (74.7)	39 (55.7)
	Regression or arrest	Progression	7 (9.3)	15 (21.4)
	Progression	Regression or arrest	7 (9.3)	6 (8.6)
	Progression	Progression	5 (6.7)	10 (14.3)
	OR 95% CI		1.00 (0.30–3.34)	2.50 (0.92–7.87)
	*p*-value		0.789	0.081
**Radiographic**	Regression or arrest	Regression or arrest	47 (62.7)	35 (50.0)
	Regression or arrest	Progression	14 (18.7)	18 (25.7)
	Progression	Regression or arrest	12 (16.0)	9 (11.3)
	Progression	Progression	2 (2.5)	8 (11.4)
	OR 95% CI		1.17 (0.50–2.76)	2.0 (0.85–5.05)
	*p*-value		0.845	0.124

McNemar test, OR: odds ratio, CI: confidence interval.

**Table 6 children-10-01164-t006:** The clinical and radiographic success (regression) vs. failure (arrest or progression of the carious lesion) outcomes for each pair at the 6- and 12-month follow-up visits.

Diagnostic Method	The Success and Failure Status in Each Pair of Lesions		Six Months N = 75 Pairs	12 Months N = 70 Pairs
	Experimental	Control	n (%) Pairs	n (%) Pairs
**Clinical (ICDAS II)**	Regression	Regression	2 (2.7)	1 (1.4)
	Regression	Arrest or progression	6 (8.0)	11 (15.7)
	Arrest or progression	Regression	0	0
	Arrest or progression	Arrest or progression	67 (89.3)	58 (82.9)
	OR∝ 95% CI		-	-
	*p*-value		0.041*	0.003*
**Radiographic**	Regression	Regression	1 (1.3)	5 (7.1)
	Regression	Arrest or progression	13 (17.3)	16 (22.9)
	Arrest or progression	Regression	4 (5.3)	6 (8.6)
	Arrest or progression	Arrest or progression	57 (76.0)	43 (61.4)
	OR 95% CI		3.25 (1.0–13.7)	2.67 (0.99–8.32)
	*p*-value		0.052	0.055

McNemar test, OR: odds ratio, CI: confidence interval, ∝ OR and CI could not be calculated, * Statistically significant (*p* < 0.05).

## Data Availability

The authors may share the data upon reasonable request.

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
