# Peer review of "Effect of Light-Curable Resin-Modified Glass Ionomer Varnish on Non-Cavitated Proximal Caries Lesions in Primary Molars: A Randomized Controlled Trial"

_children, 2023, doi:10.3390/children10071164_

Round 1

Reviewer 1 Report

This original research about the effect of light curable resin modified glass ionomer varnish on noncavitated proximal caries in primary molars is well-written and based on up-to-date literature; the text has good readability, and the paragraphs come from each other.

In general, the manuscript is interesting. Materials and methods are written completely explanatory. The results present relevant findings that are well described and discussed. The authors properly describe the limitations of the study. The conclusions are based on the results.

Nevertheless, there are some issues to review before being suitable for publication:

Title. For correction and accuracy, the term “caries” should be replaced by “caries lesions”.

Table 3. References symbols of the lowest row do not match the description in the footnote.

Table 5. OR ant CI terms must be described in the footnote for a better understanding.

Author Response

This original research about the effect of light curable resin modified glass ionomer varnish on noncavitated proximal caries in primary molars is well-written and based on up-to-date literature; the text has good readability, and the paragraphs come from each other.

In general, the manuscript is interesting. Materials and methods are written completely explanatory. The results present relevant findings that are well described and discussed. The authors properly describe the limitations of the study. The conclusions are based on the results. Nevertheless, there are some issues to review before being suitable for publication:

Thank you for your comments.

Title. For correction and accuracy, the term “caries” should be replaced by “caries lesions”.

Thank you for your comments. The word “Caries” was replaced by “Caries Lesions” in the title of the study.

The new title is:

Effect of Light Curable Resin Modified Glass Ionomer Varnish on Noncavitated Proximal Caries Lesions in Primary Molars: A Randomized Controlled Trial

Table 3. References symbols of the lowest row do not match the description in the footnote.

Thank you for noticing.

The symbols in the footnote were changed to match the symbols in Table 3 as follows:

† chi-square test

‡ Fisher Exact test

Table 5. OR ant CI terms must be described in the footnote for a better understanding.

Thank you for the comment.

A description of OR and CI was added to the footnote of Table 5 and Table 6 as follows:

OR: odds ratio

CI: confidence interval

Reviewer 2 Report

The reviewer really appreciates the efforts of the authors to conduct this study which has good clinical significance. This is a well-structured study and the manuscript is well-written without leaving any major issues. The experiment steps explained clearly in the manuscript. There are only a few corrections and suggestions from the reviewer side

Page 7 line 255 the abbreviation of DMFT needs correction.

Suggestion: if possible please add a representative clinical picture and radiograph. The reviewer believes it will increase the impact of the article. 

Author Response

The reviewer really appreciates the efforts of the authors to conduct this study which has good clinical significance. This is a well-structured study and the manuscript is well-written without leaving any major issues. The experiment steps explained clearly in the manuscript. There are only a few corrections and suggestions from the reviewer side

Thank you for your comments.

Page 7 line 255 the abbreviation of DMFT needs correction.

Thank you for the comment.

The description of the abbreviation of DMFT was written “by mistake” as for primary teeth. It was changed to permanent teeth as follows:

dmft: Decayed, missing, filled due to caries index for primary teeth.

DMFT: Decayed, missing, filled due to caries index for permanent teeth.

DI-S: The simplified debris index.

Suggestion: if possible please add a representative clinical picture and radiograph. The reviewer believes it will increase the impact of the article. 

Thank you for the suggestion.

Clinical and radiographic images were added to the manuscript as Figure 1, 3 and 4 which are added here at the end of this document.